# A Novel Contrastive Self-Supervised Learning Framework for Solving Data Imbalance in Solder Joint Defect Detection

**DOI:** 10.3390/e25020268

**Published:** 2023-01-31

**Authors:** Jing Zhou, Guang Li, Ruifeng Wang, Ruiyang Chen, Shouhua Luo

**Affiliations:** College of Biological Sciences and Medical Engineering, Southeast University, Nanjing 210096, China

**Keywords:** solder joints, defect detection, deep learning, contrastive self-supervised learning

## Abstract

Poor chip solder joints can severely affect the quality of the finished printed circuit boards (PCBs). Due to the diversity of solder joint defects and the scarcity of anomaly data, it is a challenging task to automatically and accurately detect all types of solder joint defects in the production process in real time. To address this issue, we propose a flexible framework based on contrastive self-supervised learning (CSSL). In this framework, we first design several special data augmentation approaches to generate abundant synthetic, not good (sNG) data from the normal solder joint data. Then, we develop a data filter network to distill the highest quality data from sNG data. Based on the proposed CSSL framework, a high-accuracy classifier can be obtained even when the available training data are very limited. Ablation experiments verify that the proposed method can effectively improve the ability of the classifier to learn normal solder joint (OK) features. Through comparative experiments, the classifier trained with the help of the proposed method can achieve an accuracy of 99.14% on the test set, which is better than other competitive methods. In addition, its reasoning time is less than 6 ms per chip image, which is in favor of the real-time defect detection of chip solder joints.

## 1. Introduction

Soldering chip components to PCB is an important part of printed circuit board assembly (PCBA) production [1]. Various defects occur during the soldering process of chip components [2,3]. The most common defects include missing components, excessive solder, insufficient solder, solder skips, misalignment, tombstoning, component damage, etc.

From all these representative defects shown in Figure 1, we can see that most of the defects (e.g., excessive solder, solder skips) are directly related to the solder joints and some (e.g., missing components, component damage) are not directly related to the solder joints, but they all bring some changes to the solder joints. Hence, we can detect most of the defects in the soldering process of chip components by detecting the solder joint defects. For ease of understanding, abnormal solder joint images are collectively referred to as NG images, and the normal ones are called OK images in the following parts of the paper.

Automated optical inspection (AOI) is an advanced visual detection method for PCBAs. It uses optical cameras to scan PCBAs and detects two types of failures: catastrophic failures (missing components) and quality failures (misshapen fillet or skewed components). However, due to the diversity of defect shapes, forms, and features, AOI alone cannot meet the high-precision requirements for defect detection [4]. In addition, the inspection scenarios for chip components are also complicated [5]. For example, the bending of the PCB may displace components from their original positions; blurred images or smeared surfaces of chip components can also produce false or missing positives. To solve these issues, artificial reinspection has to be involved in the production line, despite it decreasing the inspection efficiency and increasing the labor costs.

Deep learning is one of the prominent developments in artificial intelligence (AI) [6,7,8,9,10]. It has been successfully applied in various fields, such as classification, object detection, signal diagnosis, and instance segmentation [11,12,13,14,15,16]. Many deep-learning methods have been used to improve the detection accuracy of AOI for chip solder defects. Some of the methods are based on supervised learning. Cai et al. [17] used a convolutional neural network cascade to inspect solder joint defects. Hao et al. [18] used an improved Mask-RCNN to segment and classify chip soldering defects in parallel branches. Fan et al. [19] proposed an improved Faster R-CNN for PCBA solder joint defects, and components detection with a mean average precision close to 0.99. The supervised methods rely heavily on a large number of NG samples to obtain high generalization capability, but the amount and type of NG data obtained in practical production are often insufficient. Therefore, supervised methods are usually very limited in practical applications. There are many unsupervised-based studies for defect detection, which show a promising increased performance compared with conventional methods. S. Park et al. [20] proposed a residual-based auto-encoder for the defect detection of micro-LEDs (chip components) and achieved an accuracy of 95.82% on 5143 micro-LED images. Wang et al. [21] achieved an accuracy of 92.7% on 7233 solder joint images by using a generative adversarial network (GAN) [22]. F. Ulger et al. [23] proposed a new beta-variational auto-encoder (beta-VAE) architecture for the detection of solder joint defects.

Even though defect detection based on deep neural networks has been extensively explored and many related detection methods have been proposed, the detection of defects in chip solder joints is very rare. The main reasons for this situation are related to the following aspects:

1. Different components are different in internal design. It is difficult to design solder joint defect detection methods suitable for various scenarios.

2. Solder joints are the places where electronic components and PCBs are welded by hand or machine. Due to solder, time, temperature, and other factors, solder defects may take on different forms. The defective solder joint data are very few, and it is difficult to cover all types of defective solder joint data collected in the industry. Therefore, it is difficult for neural networks to extract key features that can distinguish defects from normal solder joints by using existing data.

3. As industrial detection has high requirements for accuracy and speed, many algorithms cannot meet the actual industrial requirements. Most solder joint defection still relies on machine learning and manual review, which increases human and economic costs.

In this paper, we propose a method based on CSSL to achieve a high accuracy and high efficiency of detection for the solder joint defects of chip components. Our main contributions in this paper lie in three aspects:

1. There are a few types of research on the detection of chip solder spot defects. This paper proposes a framework that can effectively detect chip solder joint defects. High-quality simulation defect data (sNG) are obtained by using data enhancement and contrastive self-supervised learning techniques. Moreover, the accuracy of detecting chip solder spot defects is improved.

2. This paper proposes three customized enhancement methods for chip defect solder joint data. A large amount of sNG data can be obtained by using OK solder joint data to make up for the lack of NG solder joint data in many scenarios. At the same time, the specific data augmentation method combined with some prior experience can better help the classifier learn the key characteristics of solder joints, to better detect defective solder joints.

3. We designed a contrastive self-supervised learning NG solder joint data filtering network (DFN) that extracted high-quality sNG data from a large volume of sNG data, ensuring that the sNG data were more closely aligned with the NG data and maximizing the value of the sNG data.

The remainder of this paper is organized as follows: in Section 2, the preliminaries of contrastive self-supervised learning are drawn; in Section 3, three data augmentation methods, the DFN, and the CSSL framework are elaborated; in Section 4, comparative experiments are conducted and results are given; in Section 5, some related issues are discussed; finally, in Section 6, the conclusion is drawn.

## 2. Preliminaries

In recent years, contrastive self-supervised learning (CSSL) has been one of the research hotspots in deep learning [24,25,26]. The main advantage is its great ability to learn semantically rich representations from numerous unlabeled data. The process of CSSL can be divided into three steps:

(1) Constructing positive or negative samples;

(2) Training the model to learn the feature similarity of these samples in the pretext task;

(3) Fine-tuning the pre-trained model in the downstream task.

A dichotomous classification illustration of the CSSL process is shown in Figure 2. In the Animals dataset, “cat” is the positive class while “not-cat” is the negative class. Assuming that negative samples are missing or very few, synthetic negative samples (s-Negative) and positive samples can be generated by some data augmentation methods. For example, s-Negative samples are created by shuffling tiles from “cat”. Positive samples are obtained by randomly rotating “cat”. The discriminator is a feature extraction network, consisting of convolutional and non-linear layers. It is used to distinguish whether the transformed data are similar to “cat”. The aim of the discriminator is to pull together semantically similar (positive) samples and push apart non-similar (negative) samples in a shared latent space. After the pretext task, the discriminator is placed on a new task for fine-tuning [27].

### 2.1. The Pretext Task

The pretext task is a kind of self-supervised learning task that learns the data representation using pseudo-labels. Most of these pseudo-labels represent some form of transformation of the data. In the pretext task, the original image acts as an anchor, and its transformed version acts as a positive sample or a negative sample. In CSSL, the pretext task design is vital to the performance of the network on the downstream task. Many studies have made substantial contributions to the design of pretext tasks [28,29,30,31,32]. He et al. [33] proposed a momentum contrastive method (MoCo) using two distinct encoders (a primary encoder and a momentum encoder) with identical architectures. Chen et al. [34] introduced a simple framework for the contrastive learning of visual representations (SimCLR), which consists of one encoder and a multi-layer perceptron (MLP) with one hidden layer. SimCLR creates pairs of transformed images to learn the similarity among them, thus maximizing the agreement between different augmentations of the same image. Many recent studies have also reported that data augmentation is important for contrastive learning [35,36]. In CSSL, the pretext task can improve the ability of networks to learn key features when using only “cat” data.

### 2.2. The Downstream Task

Downstream tasks are the actual tasks that need to be solved in CSSL. The pretext task is to use an imaginary task to strengthen the ability of the network to extract some features of data. To better evaluate this ability, pre-trained models from pretext tasks are generally used for fine-tuning to verify the downstream task. The fine-tuning process is as follows:

(1) The auxiliary task model trained by a large amount of unlabeled data is used as the pre-training.

(2) The weights of the feature extraction part of the pre-trained model are transferred to the downstream task model.

(3) The model is retrained on labeled data to obtain a model that could adapt to the new task.

The ability of self-supervised learning will also be reflected in the performance of the downstream task model.

## 3. Methods

The whole flow chart of detecting chip solder joint defects is shown in Figure 3a. In the framework, we proposed three data augmentations for generating sNG data and an sNG data filtering network for distilling sNG data.

The overall chip solder joint defect detection in the application is shown in Figure 3b. It can be divided into two steps: the first step is as indicated in the top dash-dotted rectangle, and its function it to localize the solder joints (introduced in Section 3.1); the second step is as shown in the bottom dash-dotted rectangle, and it consists of the generation and distillation of sNG images and a classification network enforced by CSSL (introduced in Section 3.2 and Section 3.3).

### 3.1. Localizing and Extracting the Chip Solder Joints

Based on the AOI platform, we can obtain complete chip component images as shown in Figure 1. To extract the chip solder joints, a multi-scale target detection network based on SSD (MobileNetV2-SSD) [37] is first introduced to localize the solder joints. The architecture of MobileNetV2-SSD is shown in Figure 4. MobileNetV2 [38] is chosen as the backbone to improve the detection speed and reduce the model complexity. In addition, two improvements have been made to meet the needs of real-time detection: (1) add three additional convolutional layers following the MobileNetV2 backbone and (2) modify the output dimension of the last convolutional layer from 256 to 64. The former increases the feature scale in the network in order to extract the feature information better. The latter reduces the number of references to reduce the network size and improve the computing speed.

### 3.2. Three Data Augmentations for Generating sNG Images

Some examples of OK and NG images are shown in Figure 5a. A magnified OK image shows its structure in Figure 5b. The right side is a complete electrode, and the left side has a uniform background because the solder is evenly distributed. Unlike OK images, NG images have irregularly shaped electrodes and various bizarre backgrounds. The detection of solder joint defects is a typical class imbalance problem [39,40]. Due to the low probability of occurrence and the diversity of solder joint defects, it is difficult to obtain comprehensive and sufficient NG samples. Hence, we empirically design three augmentation methods to obtain sNG images from OK images.

(1) *Electrode shifting.* The flow chart of the electrode shifting consists of four steps (I)-(IV), as shown in Figure 6. In step (I), the character-region awareness for text detection (CRAFT) [41] algorithm is used for localizing the electrode. CRAFT consists of an encoder and a decoder, as shown in Figure 7, in which the encoder adopts the MobileNetV2 structure. The output is a Gaussian heat map representing the electrode. In step (II), the minimum rectangle (the red rectangle in Figure 6I) enclosing the heat map is obtained by the thresholding method, and this rectangle is the electrode area. Step (II) is to separate the electrode from the background. After separating the electrode, we can obtain an electrode image $I_{S}$ and a background image $I_{C}$ with a rectangular blank area $B_{d}$ as shown in Figure 6II.

To make $I_{C}$ smooth, a process called background filling (Figure 8) in step (III) is implemented. Firstly, we randomly select a rectangular region $B_{u}$ from the area other than the blank area $B_{d}$ of $I_{C}$ and resized to the same size as $B_{d}$. Then, we use $B_{u}$ to overlay $B_{d}$ and obtain a filled image $I_{Filled}$ as shown in Figure 8b. $I_{Filled}$ can be formulated in the following equation:(1)$$I_{\mathrm{Filled}\;}\left(p\right)=I_{C}\left(p\right)+\left[1+M\left(p\right)\cdot B_{u}\left(p\right)\right],$$
where *p* is the pixel position, and M(p) is a mask matrix written as follows: (2)$$M\left(p\right)=\left\{\begin{array}{c} 1,p\in B_{d}\\ 0,p\notin B_{d} \end{array}\right.$$

Next, we set up an inpainting area $B_{z}$ (the green area in Figure 8a(b’)) by extending 5 pixels beyond the boundaries of $B_{d}$. Finally, we use the inpainting method based on fast matching (IFM) ([42]) to repair $B_{z}$ and obtain the final smooth background image $I_{IFM}$ as shown in Figure 8a(c). The specific formula is expressed as follows: (3)$$I_{IFM}\left(p\right)=\frac{\sum_{q\in B_{\varepsilon }\left(p\right)}w\left(p,q\right)\left[I_{\mathrm{Filled}\;}\left(q\right)+\nabla I_{\mathrm{Filled}\;}\left(q\right)\left(p-q\right)\right]}{\sum_{q\in B_{\varepsilon }\left(p\right)}w\left(p,q\right)}$$
where $p\in B_{z}$, $B_{\varepsilon }\left(p\right)$ means the neighborhood of *p*, and w(p,q) is a weight function of points *p* and *q* [42]. Another process in step (III) is boundary weakening, which is used to smooth the boundaries of electrode $I_{S}$. In this process, electrode $I_{s}$ is smoothed by a Gaussian kernel of the same size. The specific expression is as follows: (4)$$I_{F}\left(p\right)=\left\{\begin{array}{c} L:L_{F}\left(p\right)=L_{S}\left(p\right)\cdot M_{G}\left(p\right)\\ a:A_{F}\left(p\right)=A_{S}\left(p\right)\\ b:B_{F}\left(p\right)=B_{S}\left(p\right) \end{array}\right.$$
where $L_{S}\left(p\right)$, $A_{S}\left(p\right)$, and $B_{S}\left(p\right)$ represent the three components of the CIELAB color space of $I_{s}\left(p\right)$, and $M_{G}\left(p\right)$ is a Gaussian kernel with mean *p*. In step (IV), all generated $I_{IFM}$ and $I_{F}$ are randomly combined to generate sNG images as shown in Figure 6IV.

(2) *Random dropping.* The purpose of random dropping is to break the integrity of the electrodes as shown in Figure 9. The specific process is as follows: Firstly, resize an OK image *I* to (64,64) pixels and divide it into 16 square tiles. Secondly, select a tile $T_{c}$ from the background area to randomly replace a tile $T_{s}$ in the electrode area. After this process, we can get an image $I_{r}$ as shown in Figure 9(c1) in which part of the electrode is missing. Finally, we use the same process as in the electrode shifting to select the inpainting area $B_{z}$ and smooth the boundaries of the missing part, resulting in a final sNG image, as shown in Figure 9(c2).

(3) *Electrode morphology transformation.* Electrode morphology transformation consists of three steps as shown in Figure 10. Step (I) adopts the same boundary weakening method as in electrode shifting to smooth the electrode image $I_{S}$. In step (II), firstly, we design an elliptical mask $M_{e}\left(p\right)$ written as
(5)$$M_{e}\left(p\right)=\left\{\begin{array}{c} 1,p\in O_{e}\\ 0,p\notin O_{e} \end{array}\right.$$
where $O_{e}$ is an elliptical region with major and minor axes being the height *h* and width *w* of $I_{S}$. Then, we obtain a transformed electrode image $I_{T}$ through the following operation: (6)$$I_{T}\left(p\right)=I_{S}\left(p\right)\cdot M_{e}\left(p\right)$$

In step (III), the final sNG images can be obtained by randomly combining the transformed electrode images with the background images acquired in electrode shifting. We can use one or more of the above augmentation methods in combination to generate sNG data. Figure 11 shows some examples of the generated sNG images based on these augmentation methods. In addition, some other common data augmentations such as image rotation, image flipping, and random brightness can also be combined in practice.

### 3.3. Construction of DFN

To produce sNG data closer to NG data, a DFN is proposed to distill the best sNG data from the sNG data directly obtained by the augmentation methods. Because of the diversity and scarcity of NG data, to construct a DFN based on supervised learning is not easy to achieve, so we build a DFN based on contrastive self-supervised learning. The proposed DFN is shown in Figure 12, and it consists of three modules: data augmentation, encoder, and projection. In the data augmentation module, two augmented images $T_{1}$ and $T_{2}$ are obtained from an NG image $T_{0}$ through two separate random augmentations. The data augmentations employed include rotation, cropping, resizing, flipping, cutout, erasing, and noise injection. The encoder is based on MobileNetV2, and it can map $T_{n}\left(n=0,1,2\right)$ to a vector $h_{n}$ in the latent space. In the latent space, when the DFN is fully trained, the core features of each type of NG data can be extracted, which can be used as a good metric to evaluate the similarity between NG data and sNG data. To train NG, a vector $h_{n}$ in the latent space is transferred to a vector $z_{n}$ in the contrastive space through a multi-layer perceptron, which consists of two fully connected layers and a ReLU activation layer between them. Assuming the batch size of the training data is *N*, the loss function $L_{DFN}$ is denoted as
(7)$$L_{DFN}=\frac{1}{2N}\sum_{k=1}^{N}\left[l_{WNT-Xent}\left(z_{2}^{k},z_{1}^{k},z_{0}^{k}\right)+l_{WNT-\;\mathrm{Xent}\;}\left(z_{1}^{k},z_{2}^{k},z_{0}^{k}\right)\right]$$
where $l_{\mathrm{WNT}-\mathrm{Xent}\;}$ is the weighted sum of multiple normalized temperature-scaled cross entropy (NT-Xent) functions: (8)$$\begin{array}{c} l_{WNT-Xent}\left(z_{i}^{k},z_{j}^{k},z_{0}^{k}\right)=\lambda l_{NT-Xent}\left(z_{i}^{k},z_{j}^{k}\right)+\frac{1}{4}\left(1-\lambda \right)\left[l_{NT-Xent}\left(z_{i}^{k},z_{0}^{k}\right)+\right.\\ \left.l_{NT-Xent}\left(z_{0}^{k},z_{i}^{k}\right)+l_{NT-Xent}\left(z_{j}^{k},z_{0}^{k}\right)+l_{NT-Xent}\left(z_{0}^{k},z_{j}^{k}\right)\right] \end{array}$$ The NT-Xent function is expressed as follows: (9)$$l_{NT-Xent}\left(z_{i}^{k},z_{j}^{k}\right)=-\log\left(\frac{\exp\left(\frac{\mathrm{sim}\left(z_{i}^{k},z_{j}^{k}\right)}{\tau }\right)}{\sum_{m=1}^{N}\;L_{\left[m\neq k\right]}\exp\left(\frac{\mathrm{sim}\left(z_{i}^{k},z_{1}^{m}\right)}{\tau }\right)+\sum_{m=1}^{N}\;L_{\left[m\neq k\right]}\exp\left(\frac{\mathrm{sim}\left(z_{i}^{k},z_{2}^{m}\right)}{\tau }\right)}\right),$$
where $L_{\left[m\neq k\right]}\in \left\{0,1\right\}$ is an indicator function that is equal to 1 if m≠k and 0 otherwise, τ is a temperature parameter, and the function sim is a distance measure of two vectors written as
(10)$$\mathrm{sim}\left(z_{i}^{k},z_{j}^{k}\right)=\frac{{\left(z_{i}^{k}\right)}^{T}z_{j}^{k}}{∥z_{i}^{k}∥∥z_{j}^{k}∥}$$

### 3.4. Detection Model Based on the CSSL Framework

The whole flow chart of the detection model based on the CSSL framework is shown in Figure 13, and it consists of three parts. The first two parts are introduced in Section 3.1 and Section 3.2. After processing these two parts, OK data, NG data, and sNG data are prepared. Based on these data, we can build a detection framework based on self-supervised learning (SSL) or contrastive self-supervised learning (CSSL). In the SSL framework, sNG images and OK images are directly used to pre-train a classification network, and then the pre-trained network is transferred to the downstream classification task, and finally, OK images and a small amount of NG images are used to fine-tune the pre-trained network. Since the DFN is trained, each NG image $T^{k}$ is mapped to a vector $z^{k}$ in the contrastive space, and we can measure the average distance between each sNG image $S^{m}$ whose corresponding vector is $z_{s}^{m}$ and all NG images in the contrastive space by the following formula: (11)$$D_{m}=\frac{1}{N}\sum_{k=1}^{N}-\log\left(\exp\left(\frac{z^{k}\cdot z_{s}^{m}}{\left|z^{k}\right|\left|z_{s}^{m}\right|}\right)\right)$$ Here, the smaller $D_{m}$ means the better similarity. Therefore, we can sort the sNG images according to the $D_{m}$ value of each sNG image and then distill the best sNG images. The steps in the CSSL framework after the distilling step are the same as those in the SSL framework.

## 4. Datasets and Experiments

In this section, comparative experiments were conducted to demonstrate the performance of the CSSL method based on the specifically augmented data and the proposed DFN model. The experiments were performed on a graphics station equipped with an Nvidia GTX 1080Ti GPU and 16 GB of RAM. The trained network was finally deployed on a computer with an Intel Core i5-7300HQ CPU with 8 GB of RAM memory in the production line for real-time detection.

### 4.1. Dataset Preparation

There were two sets of chip component data in this study from two electronic manufacturing companies, provided by Shenzhen MagicRay Technology Co., LTD. Based on these two sets of data, we used the MobileNetV2-SSD method to extract OK and NG data to establish two datasets, Chip-A and Chip-B. The details of the two datasets are shown in Table 1.

(1) Chip-A consisted of 26,521 real solder joint images (23,224 OK images and 3297 NG images) and 21,372 sNG images. All real images were collected by professionals. The NG set consisted of 1031 images of excessive solder, 429 images of insufficient solder, 403 images of missing components, 634 images of component damage, and 800 images of misalignment. The sNG images were generated from OK images by the three specific augmentations. In the pretext tasks, 10,000 OK images and 10,000 sNG images were combined into the training set, and the test set contained 5000 OK images, 1037 NG images, and 5000 sNG images. In the downstream tasks, the training set contained 5244 OK images and 852 NG images, and the test set consisted of 3000 OK images and 1408 NG images. In the supervised learning, the training set consisted of 20,224 OK images and 1889 NG images, and the test set was the same as in the downstream tasks.

(2) Chip-B consisted of 20,512 real solder joint images (18,600 OK images and 1912 NG images) and 17,200 sNG images. In the pretext tasks, the training set consisted of 8000 OK images and 8000 sNG images, and the test set contained 4000 OK images, 592 NG images, and 4000 sNG images. In the downstream tasks, the training set consisted of 4000 OK images, 823 NG images, and 2600 sNG images. The test set contained 2600 OK images, 832 NG images, and 2600 sNG images. In the supervised learning, the training set consisted of 13,020 OK images and 1089 NG images, and the test set contained the same OK and NG images as in the downstream tasks.

### 4.2. Evaluation Metrics

To evaluate the performance of the proposed method, three metrics including precision (*P*), recall (*R*), and *F*1-score (*F*1) were used in this study. The confusion matrix is shown in Table 2. *P*, *R*, and F1 are defined as follows: (12)$$P=\frac{TP}{TP+FP}$$
(13)$$R=\frac{TP}{TP+FN}$$
(14)$$F1=2\times \frac{P\times R}{P+R}$$

### 4.3. Real Experiments

In this subsection, three experiments were conducted. Firstly, we performed a comparative experiment of supervised learning (SL) methods and SSL methods. Then, we performed an experiment to show the performance improvement of the CSSL method compared to the SSL method. Finally, we compared the CSSL method with the state-of-the-art defect detection methods.

#### 4.3.1. Comparative Experiments of SL Methods and SSL Methods

To better demonstrate that SSL methods have better detection accuracy than SL methods when training data are insufficient, we introduced two classic classifiers, VGG16 [43] and MobileNetV2. Based on these two classifiers, we built two supervised models named VGG16-SL and MobileNetV2-SL and two self-supervised models named VGG16-SSL and MobileNetV2-SSL. When training the SL models, the initial learning rate was set to 0.001, the Adam algorithm [44] was used as the optimizer, and the loss function was categorical cross-entropy. Both SL models were trained for 2000 epochs. NG data were augmented by several common data augmentation methods, including rotation, flipping, and noise injection. When training the SSL models, the initial learning rate for the Adam optimizer was set to 0.001, and both SSL models were trained for 5000 epochs in the pretext tasks. After the pre-training was finished, the pre-trained models were transferred to the downstream tasks for fine-tuning, and the learning rate was 0.0002. For the sake of fairness, the common data augmentations applied to the NG data were also used for the sNG data.

The comparison results are shown in Table 3. From the F1-scores in the table, we can see that VGG16-SSL outperforms VGG16-SL, and MobileNetV2-SSL outperforms MobileNetV2-SL. These two results are consistent, and they can demonstrate that SSL methods can achieve better defect detection accuracy than SL methods. It is worth noting that the amount of the NG data used to train the SSL models is much smaller than the amount of NG data to train the SL models. This demonstrates that SSL models are more suitable for the scenario of insufficient NG data than SL models. By comparing the values of all metrics in Table 3, we can also see that MobileNetV2-SSL outperforms VGG16-SSL. To validate the performance in real applications, the fully trained VGG16-SSL and MobileNetV2-SSL were deployed in the production line. The parameter number and inference speed for both models are shown in Table 4. It can be seen that MobileNetV2-SSL has fewer parameters and a faster inference speed than VGG16-SSL. Summarizing the above results, we can conclude that MobileNetV2 has better detection accuracy, better efficiency, and fewer parameters than VGG16; therefore, in the following comparative experiments, we use MobileNetV2 as the classifier of the CSSL Framework.

#### 4.3.2. Validation of the CSSL Framework

To further improve the detection accuracy of MobileNetV2-SSL, a contrastive self-supervised learning network is added to MobileNetV2-SSL to construct a CSSL framework. Unlike the traditional CSSL networks in computer vision, which are directly used in the classification process, our CSSL network is used to distill high-quality sNGs before the pretext tasks. Specifically, first, we generated 100,000 sNGs by the three special augmentations above. Then, we calculated the average similarity value between each sNG image and all NG images through the trained DFN. The output similarity values of all sNG images were in the range $S_{sim}=\left(8.02,102.47\right)$. After that, we sorted sNG images according to the similarity value. To better validate the effectiveness of the DFN, we set 4 distillation rates (r=0.25,0.5,0.75,1.0) to distill sNG images, meaning the top r×100 percent of the sNG images ordered by the similarity value were extracted. For simplicity and better understanding, we referred to MobileNetV2-SSL incorporating DFN with a distillation rate *r* as *r*-MobileNetV2-CSSL. For example, 0.25-MobileNetV2-CSSL is a materialized model under the CSSL framework with the distillation rate being 0.25 and the classifier being MobileNetV2. In all models, 5000 epochs were trained for the pretext tasks and 1000 epochs for the downstream tasks. Other experimental parameters were set to be the same as those used in the first experiment above.

The comparison results of the four models are shown in Table 5. From the F1-scores on both datasets, we can see that the performance of *r*-MobileNetV2-CSSL improves as *r* decreases, and 0.25-MobileNetV2-CSSL achieves the best F1-scores. These results validate that DFN can effectively distill the best sNG images to help the network improve detection accuracy. To validate the performance of the CSSL framework in the production line, an additional 2078 images of chip solder joints with defects were collected from the production line for testing 0.25-MobileNetV2-CSSL. The detection results are shown in Table 6, and it is clear that the detection accuracy is very high and can meet the accuracy needs in practical applications.

#### 4.3.3. Performance Comparison of the CSSL Framework with Other Classic Methods

To more intuitively validate the performance of our CSSL framework, several state-of-the-art defect inspection methods such as DIM [45], AMDIM [46], CPC-v2 [47], GANomaly [22], and GSDD [48] were introduced for comparison. The best-performing model in the above second experiment, 0.25-MobileNetV2-CSSL, was used as the comparison model for these competing methods. The experimental parameter settings of 0.25-MobileNetV2-CSSL were the same as in the second experiment above.

The comparison results are shown in Table 7. It can be seen that GANomaly has the highest *p* value on the TOK data and the CPC-v2 has the highest *p* value on the NG data. However, GANomaly has a relatively low *p* value on the NG data, which means the detection performance of GANomaly on NG images is not good. Part of the reason is that there is not enough NG information to support the training process of GANomaly. Most of the *p* values and R values of CPC-v2 and DIM are comparably good except for the R value on the NG data. CPC-v2 learns features by extracting features from each block by dividing the OK image into blocks. Therefore, CPC-v2 is more sensitive to OK data, while it is unfamiliar to NG data that it has never met. DIM constructs contrastive learning tasks by local features in OK images. Its discrimination ability for OK data is also much better than that for NG data. For chip solder joint defect detection, we should pay more attention to the recall rate of NG data. From the perspective of industrial production, if NG data are missed, it will cause more potential risks. Therefore, DIM and CPC do not meet the ideal requirements for solder joint defect detection. The proposed method is compared with a similar research method GSDD, and it is found that GSDD can not cover more complex defect scenes by solving the problem of defect classification through the color gradient. F1-score is a comprehensive metric that can better evaluate the ability of a method in detection accuracy. From the F1 values in Table 7, we can see that 0.25-MobileNetV2-CSSL outperforms all the competing methods. In particular, 0.25-MobileNetV2-CSSL makes a significant improvement to the F1-score on the NG data compared to all competing methods.

### 4.4. Ablation Study

#### 4.4.1. Effectiveness of the Three Specifically Designed Data Augmentation Methods

To demonstrate the contribution of each augmentation method, we performed an ablation study. Most of the experiment settings are the same as in the second experiment above, except that the three augmentation methods are optional during the training of the DFN. The results of the ablation study are summarized in Table 8. In this table, *a*, *b*, and *c* represent electrode shift, random dropping, and electrode morphology transformation, respectively. It can be seen that any combination of two augmentation methods is better than a single augmentation method in terms of the F1-score. By comparing all three combinations of two augmentation methods, we can see that the combination of the three augmentations has the best performance.

#### 4.4.2. Effectiveness of WNT-Xent Loss

To demonstrate the effectiveness of the WNT-Xent, we compared four contrastive loss functions (NT-Xent, NT-Logistic, InfoNCE, and Margin Triplet). Table 9 shows the results of the contrastive loss function ablation study. It can be seen that WNT-Xent performs best for distilling sNG images compared to other contrastive loss functions.

## 5. Discussion

Supervised learning is limited by the requirement for a large amount of labeled data in many application scenarios. Solder joint defect detection is one such scenario where labeled NG data are so insufficient; hence, it is a typical class imbalance problem. Usually, we can obtain a large amount of normal data but very little abnormal data. If we use such a dataset to train a supervised network, there is a high probability that the trained network will perform poorly in identifying abnormal data. The potential approach to solve this issue is to augment abnormal data to make as much of them as normal data. However, the common augmentation methods are not suitable for this case, because augmenting data from NG samples through these common methods can only generate similar defect images to the existing ones in the training set. Therefore, we propose several augmentation methods based on the OK data. The benefit of this idea is that we can transform, reshape, or modify a large number of normal images to simulate abnormal images. Due to a large amount of OK data and a random combination of various transformations that simulate real defects, we can obtain a great number of sNG images, many of which are similar to NG data. In the first comparative experiment, we have shown that these sNG images are very helpful for improving defect detection accuracy (Table 8). In the univariate experiments, shifting is better than random missing. Electrode morphological transformation is better than electrode shifting. This indicates electrode morphology is the key feature that can be extracted from MobileNetV2 to distinguish OK and NG solder joint data. In the multivariate experiments, the combination of the three data transformations performs best. This result verifies from the side that the generated sNG data by our proposed augmentation methods can simulate the NG data well. It is worth noting that we propose only three augmentation methods, but there should be many other augmentation methods that can be used to further improve detection accuracy or solve some other less common detection problems.

The data filtering network DFN proposed in the paper is a very important support to the whole framework, because in the augmentation process we can only generate a great amount of NG data without prioritization, and we do not know which NG image is more helpful to the training process. The DFN is based on contrastive self-supervised learning, and it can be trained by using a little NG data. After training, the core feature vector of each NG image or sNG image can be extracted through the DFN, so that high-quality sNG images can be extracted according to the average distance between the feature vector of each sNG image and the feature vectors of all NG images. In the experiments, we can see that the detection accuracy increases as the distillation rate decreases, which is good proof that the DFN successfully extracts the sNG data that best match the NG data. The benefit of DFN in the paper is that we can select good samples and discard bad samples in a targeted manner, thereby effectively improving the quality of the enhanced data. By doing so, we do not need to worry about the negative impact of wrongly generated sNG data on the classification network any longer.

On the basis of the special augmentation methods and the DFN, we develop a CSSL framework. The reason we call it a framework is that it is like a frame with many replaceable or adjustable modules. For example, we can customize the augmentation methods, select an appropriate classifier, set a suitable distillation rate for the DFN, and even further optimize the DFN to better sift the sNG data. In the comparative experiment with several state-of-the-art self-supervised methods and unsupervised methods, we find that it is not a complicated thing to set up a model under the framework that can achieve better performance than these classical methods. This demonstrates that the proposed framework still has a lot of room for improvement. Moreover, when this model is deployed in the production line, it is found that it can achieve an accuracy of 99.14% with an average detection speed of about 6 ms per chip image. It is a satisfactory achievement for the practical application.

Although our method has not been thoroughly tested on various datasets with large amounts of defect data, we are confident that the proposed framework will be suitable for various application scenarios since the framework is very flexible. However, we may need to design other special augmentation methods, develop better classification networks, or further optimize the DFN to make the framework adapt to some uncommon situations.

## 6. Conclusions

In this paper, we propose three specifically designed augmentation methods to generate sNG data and build a DFN to select the best sNG data. Based on these special augmentation methods and the DFN, we construct a CSSL framework and validate its effectiveness through a series of comparative experiments. Experiments show that the CSSL framework can achieve better detection accuracy than the supervised methods and the state-of-the-art unsupervised and self-supervised methods in the case of insufficient NG data. An optimized model based on the CSSL framework has been deployed in the production line and can achieve an accuracy of 99.14%. The average detection speed is about 6 ms per chip image, which can meet the real-time detection needs in applications.

Nevertheless, there is still much room for improvement regarding the research in this paper. There are two aspects, as follows:

(1) The detection of some uncommon solder joint defects still needs further research.

(2) The framework proposed in this paper can further simplify the process of training classifiers and save training time.

In the future, we will focus on solving the above problems and extend the proposed framework to as many components as possible.

## Figures and Tables

**Figure 1 entropy-25-00268-f001:**
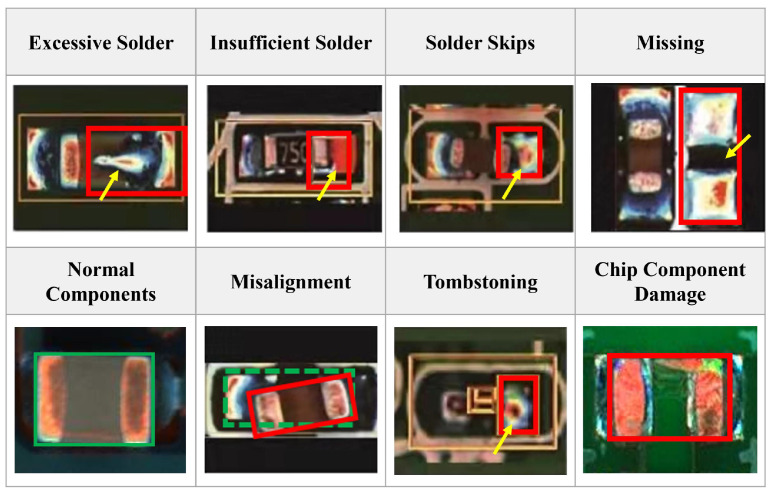
Several common defects occurring in soldering process of chip components. The solid red box is the defective area, and the green box is the correct soldering area for the component. The yellow arrow points to the defect curve.

**Figure 2 entropy-25-00268-f002:**
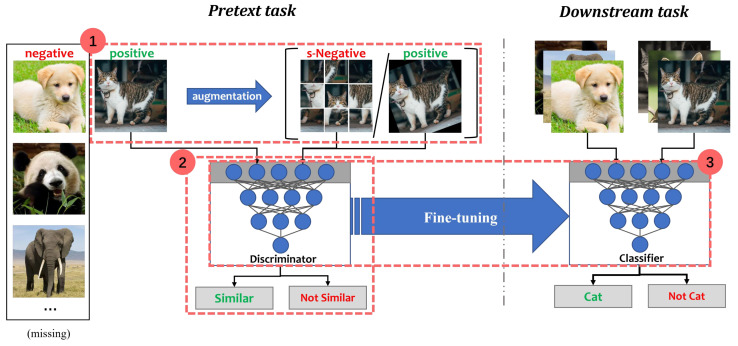
Contrastive self-supervised learning: (1) constructing positive or negative samples; (2) training the model to learn the feature similarity of these samples in the pretext task; (3) fine-tuning the pre-trained model in the downstream task.

**Figure 3 entropy-25-00268-f003:**
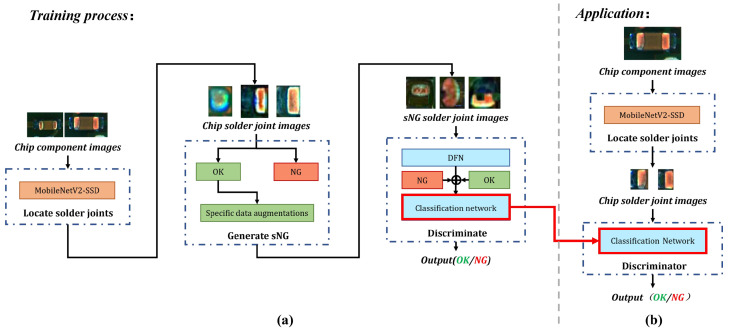
(**a**) The overall flow chart of the proposed framework; (**b**) the flow of chip solder joint defect detection in application. The red solid arrows represent the classifier model deployment.

**Figure 4 entropy-25-00268-f004:**
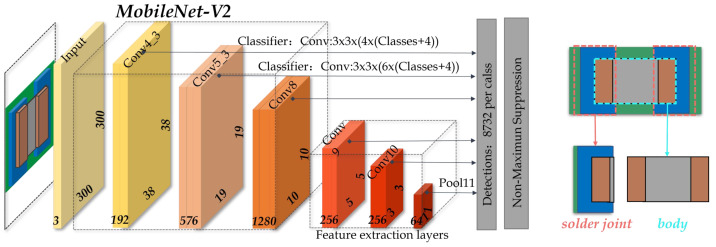
The structure of MobileNetV2-SSD.

**Figure 5 entropy-25-00268-f005:**
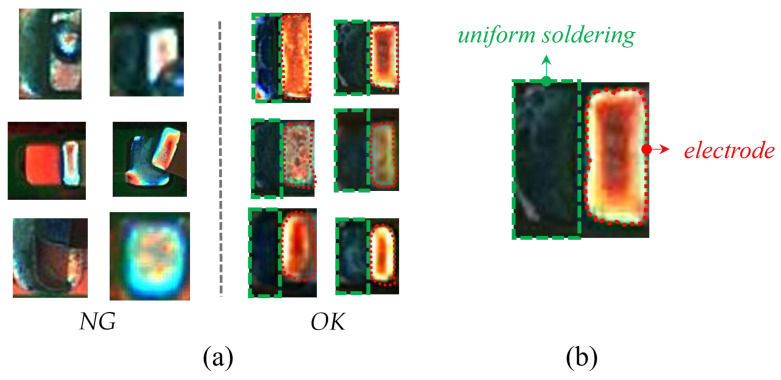
(**a**) is the comparison of NG and OK samples, and (**b**) is a magnified OK image with uniform solder on the left and a complete electrode on the right. The dashed green area indicates the solder, and the curved red area indicates the electrode.

**Figure 6 entropy-25-00268-f006:**
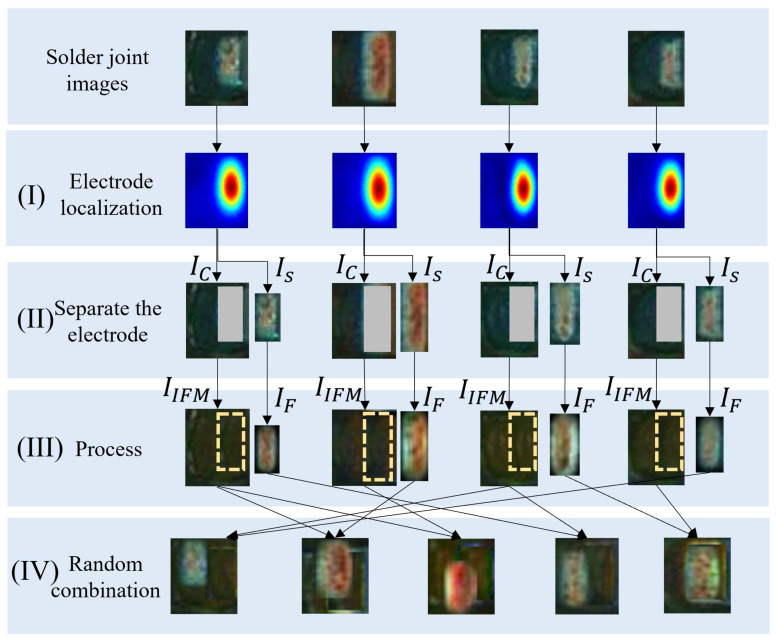
The flow chart of electrode shifting: (**I**) is to localize the electrode; (**II**) is to separate the electrode; (**III**) is to smooth the boundaries of the electrode and fill the background (uniform soldering); (**IV**) is the random combination of the transformed electrodes and the background images acquired in electrode shifting.

**Figure 7 entropy-25-00268-f007:**
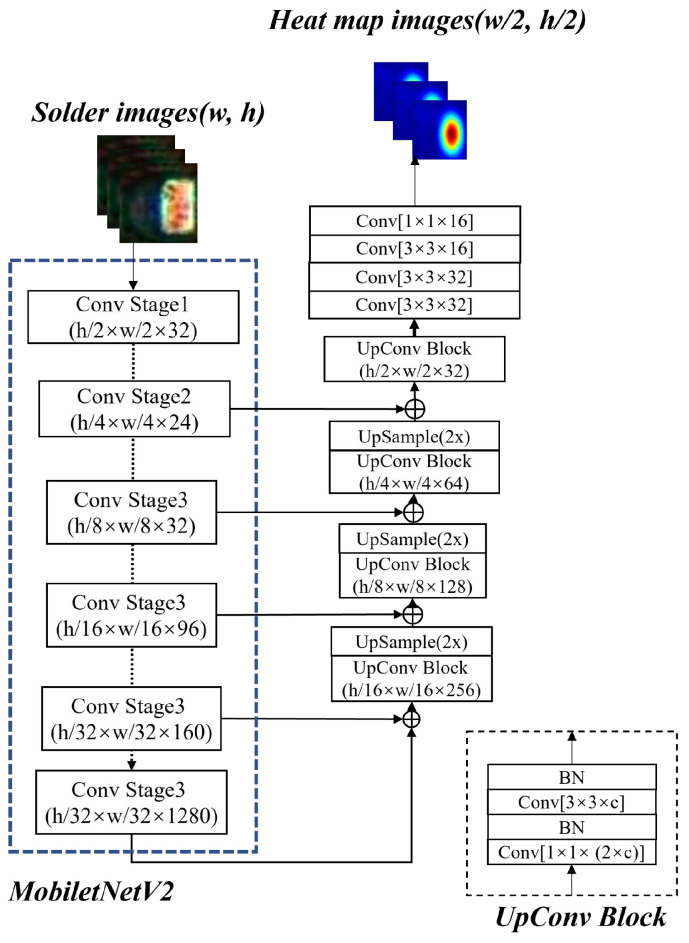
The structure of the CRAFT used to obtain the Gaussian heat map indicating the electrode. The left part is the encoder, which is based on MobileNetV2, and the right part is the decoder, which has the same structure as the original CRAFT.

**Figure 8 entropy-25-00268-f008:**
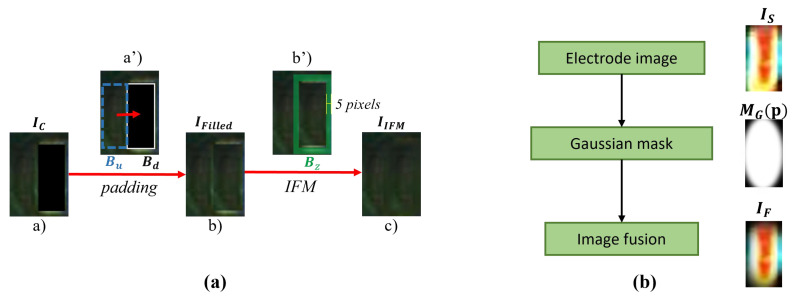
(**a**) is the background filling process in which (a’) is the solder joint image with electrodes removed, (b’) is the filled image, and (c): is the final smooth background image repaired by IFM. (**b**) is the boundary weakening process.

**Figure 9 entropy-25-00268-f009:**
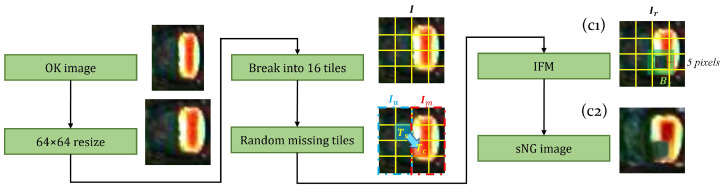
The diagram of random dropping. The IFM action area is green translucent. (c1) is the solder joint image with a part of the electrode missing, and (c2) is the final smooth background image repaired by IFM.

**Figure 10 entropy-25-00268-f010:**
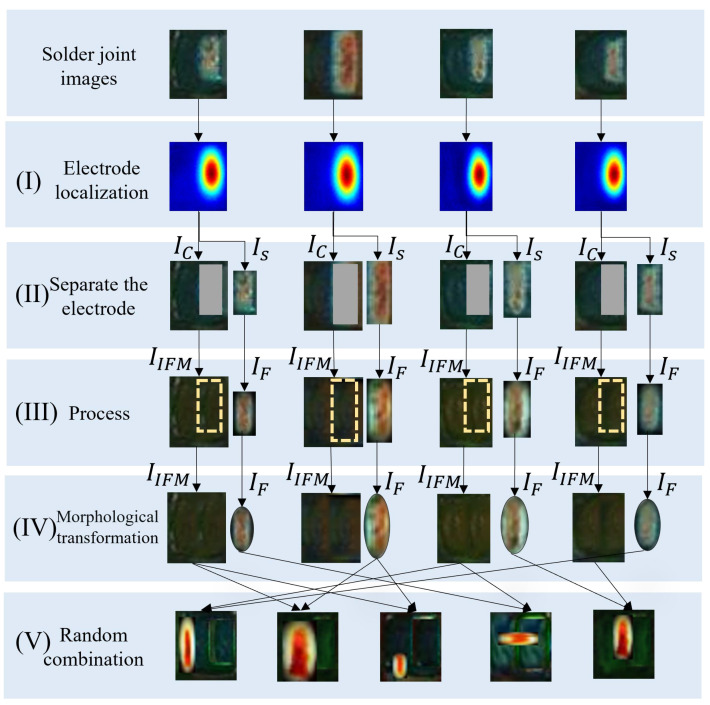
The steps of electrode morphology transformation: (**I**) is to localize the electrode; (**II**) is to separate the electrode; (**III**) is to smooth the boundaries of the electrode and fill the background (uniform soldering); (**IV**) is to transform the shape of the electrode; (**V**) is the random combination of the transformed electrodes and the background images acquired in electrode shifting.

**Figure 11 entropy-25-00268-f011:**
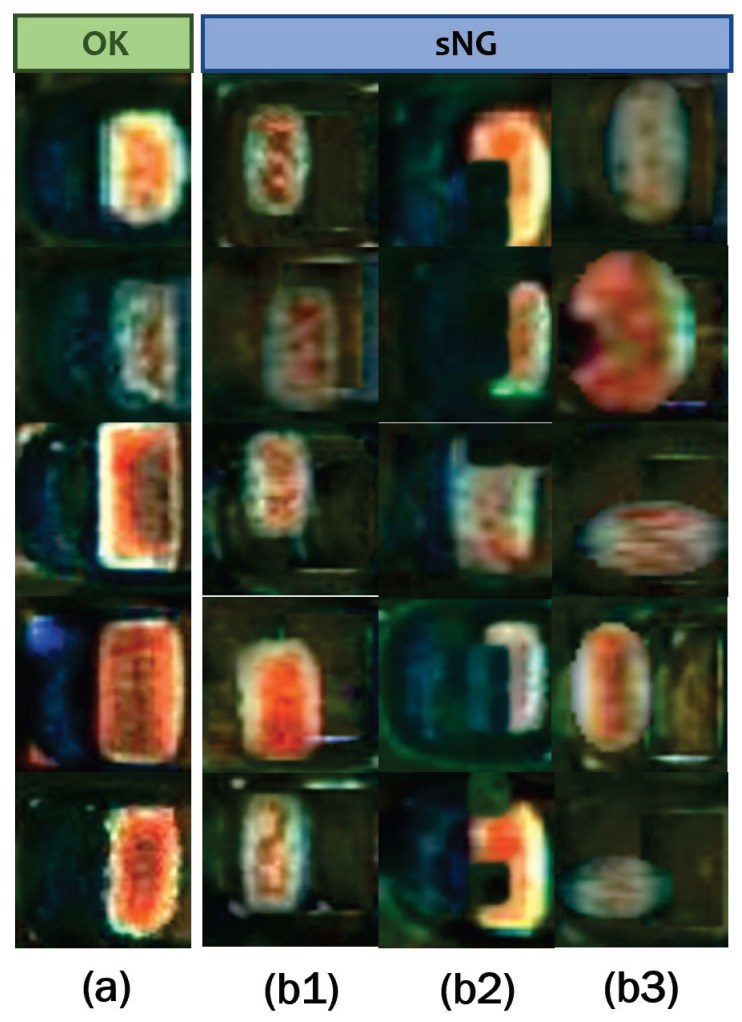
sNG data by the three proposed augmentation methods based on OK data: (**a**) OK data; (**b1**) NG data obtained through electrode shifting; (**b2**) NG data obtained through random dropping; (**b3**) NG data obtained through electrode morphology transformation.

**Figure 12 entropy-25-00268-f012:**
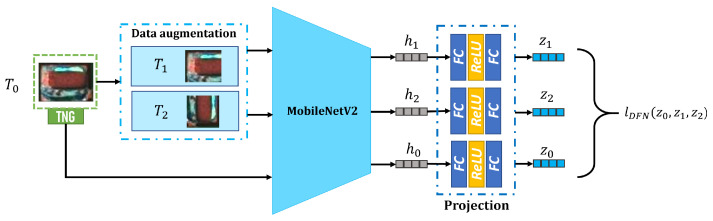
The structure of DFN.

**Figure 13 entropy-25-00268-f013:**
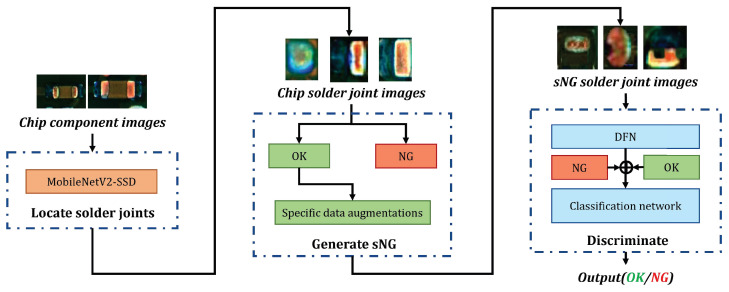
The overall flow chart of the CSSL framework.

**Table 1 entropy-25-00268-t001:** The data division of contrastive self-supervised learning and supervised learning.

Method		Chip-A Dataset						Chip-B Dataset					
		Training Set			Test Set			Training Set			Test Set		
		TOK	NG	sNG	OK	NG	sNG	OK	NG	sNG	OK	NG	sNG
CSSL	Pretext	10,000	0	10,000	5000	1640	5000	8000	0	8000	4000	956	4000
	Downstream	5224	852	3372	3000	1408	3000	4000	497	2600	2600	832	2600
SL		20,224	1889	0	3000	1408	0	13,020	1089	0	2600	832	0

**Table 2 entropy-25-00268-t002:** Calculation of Precision and Recall in the confusion matrix.

		Prediction	
		Positive	Negative
Label	Positive	TP	FP
	Negative	FN	TN

**Table 3 entropy-25-00268-t003:** Supervised learning and contrastive self-supervised learning.

Classifier	Chip-A Dataset						Chip-B Dataset					
	Recall		Precision		F1-Score		Recall		Precision		F1-Score	
	OK	NG	OK	NG	OK	NG	OK	NG	OK	NG	OK	NG
VGG16-SL	0.8993	0.7619	0.9323	0.6748	0.9155	0.7157	0.9195	0.7454	0.8798	0.8203	0.8992	0.7811
MobileNetV2-SL	0.9427	0.7690	0.9006	0.8581	0.9212	0.8111	0.9297	0.8406	0.8778	0.9066	0.9029	0.8724
VGG16-CSSL	0.9499	0.7987	0.9116	0.8796	0.9304	0.8372	0.9597	0.9019	0.9282	0.9442	0.9437	0.9226
MobileNetV2-CSSL	0.9114	0.8840	0.9515	0.8011	0.9310	0.8405	0.9650	0.9607	0.9607	0.9649	0.9628	0.9627

**Table 4 entropy-25-00268-t004:** Comparison of VGG16 and MoblieNet-V2 parameters and inference speed.

Model	Params	Inference Speed
VGG16	183 M	19 ms/chip
MobileNetV2	3.47 M	6 ms/chip

**Table 5 entropy-25-00268-t005:** Detection results of MobileNetV2 with DFN. 0.25, 0.5, and 0.75 represent the distillation rates of the top 25%, top 50%, and top 75%.

Method	Chip-A Dataset						Chip-B Dataset					
	Recall		Precision		F1-Score		Recall		Precision		F1-Score	
	OK	NG	OK	NG	OK	NG	OK	NG	OK	NG	OK	NG
0.25-DFN-MobileNetV2	0.9826	0.9919	0.9972	0.9517	0.9898	0.9714	0.9896	0.9861	0.9952	0.9704	0.9924	0.9782
0.5-DFN-MobileNetV2	0.9930	0.9351	0.9777	0.9789	0.9853	0.9564	0.9801	0.9756	0.9755	0.9802	0.9778	0.9779
0.75-DFN-MobileNetV2	0.9222	0.9381	0.9638	0.8709	0.9425	0.9032	0.9732	0.9686	0.9686	0.9732	0.9709	0.9709
1.0-DFN-MobileNetV2	0.9114	0.8840	0.9515	0.8011	0.9310	0.8405	0.9650	0.9607	0.9607	0.9649	0.9628	0.9627

**Table 6 entropy-25-00268-t006:** Detection accuracy of chip solder joint defects.

Defect Type	Number	Accuracy
Excessive solder	842	99.20%
Insufficient solder	523	98.37%
Missing or wrong components	249	100%
Misalignment	371	98.34%
Chip component damage	93	99.78%
Total	2078	99.14%

**Table 7 entropy-25-00268-t007:** Detection results of MobileNetV2 based on our proposed method.

Method	Classifier	Chip-A Dataset						Chip-B Dataset					
		Recall		Precision		F1-Score		Recall		Precision		F1-Score	
		OK	NG	OK	NG	OK	NG	OK	NG	OK	NG	OK	NG
DIM [45]	MobileNetV2	0.9920	0.9208	0.9731	0.9755	0.9825	0.9474	0.9879	0.9494	0.9812	0.9672	0.9845	0.9582
AMDIM [46]		0.9310	0.9925	0.9972	0.8335	0.9629	0.9061	0.9532	0.9878	0.9958	0.8737	0.9740	0.9273
CPC v2 [47]		0.9930	0.9351	0.9777	0.9789	0.9853	0.9564	0.9932	0.9310	0.9764	0.9794	0.9558	0.8909
GANomaly [22]		0.9120	0.9960	0.9985	0.7966	0.9533	0.8852	0.9168	0.9954	0.9983	0.8963	0.9558	0.8909
GSDD [48]		0.9818	0.9342	0.9370	0.9809	0.9589	0.9570	0.9831	0.9387	0.9439	0.9387	0.9631	0.9596
0.25-DFN-MobileNetV2		0.9826	0.9919	0.9972	0.9517	0.9898	0.9714	0.9896	0.9861	0.9952	0.9704	0.9924	0.9782

**Table 8 entropy-25-00268-t008:** Ablation study of specific data augmentations. (a) Electrode shift; (b) random missing; (c) electrode morphology transformation. “↑”: The highest result of the double transform experiment is higher than that of the single transform experiment with *a*. “↓”: The highest result of the double transform experiment is lower than that of the single transform experiment with *a*.

Method	a	b	c	Recall		Precision		F1-Score	
				OK	NG	OK	NG	OK	NG
MobileNet-V2	√	-	-	0.9955	0.7607	0.9218	0.9836	0.9572	0.8579
	-	√	-	0.9938	0.2381	0.7904	0.9300	0.8805	0.3792
	-	-	√	0.9992	0.8052	0.9368	0.9971	0.9670	0.8909
	√	√	-	0.9908	0.8052	0.9363	0.9680	0.9628	0.8791
	√	-	√	0.9986	0.8687	0.9565	0.9954	0.9532	0.9101
	-	√	√	0.9922	0.7688	0.9254	0.9715	0.9576	0.8583
	√	√	√	0.9780 (−0.0175)↓	0.8821 (+0.1214)↑	0.9600 (+0.0382)↑	0.9328 (−0.0510)↓	0.9689 (+0.0082)↑	0.9224 (+0.0652)↑

**Table 9 entropy-25-00268-t009:** Ablation study of loss functions.

DFN Loss	Classifier	Recall		Precision		F1-Score	
		OK	NG	OK	NG	OK	NG
NT-Xent(baseline)	MobileNetV2	0.9028	0.9287	0.8789	0.9435	0.8907	0.9360
NT-Logistic		0.9491	0.9074	0.9444	0.9148	0.9468	0.9111
InfoNCE		0.9709	0.8990	0.9231	0.9612	0.9464	0.9291
Margin Triplet		0.9477	0.9188	0.9184	0.9480	0.9328	0.9332
WNT-Xent(Ours)		0.9884	0.9185	0.9784	0.9549	0.9834	0.9364

## Data Availability

Not applicable.

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
