# Peer review of "A Novel Contrastive Self-Supervised Learning Framework for Solving Data Imbalance in Solder Joint Defect Detection"

_entropy, 2023, doi:10.3390/e25020268_

Round 1

Reviewer 1 Report

Poor chip solder joints can severely affect the quality of the finished PCBs. Due to the diversity of solder joint defects and the scarcity of anomaly data, it is a challenging task to automatically and accurately detect all types of solder joint defects in the production process in real time. To address this issue, we propose a flexible framework based on contrastive self-supervised learning (CSSL). In this framework, we first design several special data augmentation approaches to generate abundant synthetic, not good (sNG) data from the normal solder joint data. Then, we develop a data filter network to distill the highest quality data from sNG data. Based on the proposed CSSL framework, a high-accuracy classifier can be obtained even when the available training data is very limited. Experiments show that with the help of the proposed CSSL framework, the trained classifier can achieve an accuracy of 99.14% on the test set, outperforming the competing methods. In addition, its reasoning time is less than 6ms per chip image, which is in favor of real-time defect detection of chip solder joints. This is an interesting research paper. There are some suggestions for revision.

1)       The motivation is not clear. Please specify the importance of the proposed solution.

2)       In introduction, the listed contributions are a little bit weak. Please highlight the innovations of the proposed solution.

3)       In the Section 2.2, the thesis mentions that “Hence, we empirically design three augmentation methods to obtain sNG images from OK images.” Is the empirical setting mentioned here based on facts?

4)       At the end of the abstract, it will be more intuitive and convincing to illustrate the qualitative results of a large number of experiments for verifying the superiority and effectiveness.

5)       In the Section 2.3, the thesis mentions that “In the latent space, when the DFN is fully trained, the core features of each type of NG data can be extracted, which can be used as a good metric to evaluate the similarity between NG data and sNG data.” What are the core features here?

6)       In the Section 2.3, Why are there no speed related indicators in the evaluation indicators?

7)       In the reference section, it will be better to search and cite more latest research, (such as "Deep residual networks-based intelligent fault diagnosis method of planetary gearboxes in cloud environments", Simulation Modelling Practice and Theory 116, 102469, 2022), which can better reflect the innovation of this thesis.

8)       Please discuss how to obtain the suitable parameter values used in the proposed solution.

9)       The experimental results are not convincing. Please compare the proposed solution with more recently published solutions.

10)    Make sure your conclusions appropriately reflect on the strengths and weaknesses of your work, how others in the field can benefit from it, and thoroughly discuss future work.

11)    The lightweight of the model is not considered, which is not conducive to embedded deployment.

Reviewer 2 Report

A framework based on contrastive self-supervised learning to detect solder joint defects is proposed.

In the introduction, the authors should explain in a clearer way what are the critical points of the methods proposed in the recent literature of solder joint defects detection and what are the objectives of the research. It is not enough for the authors to highlight the importance of data augmentation for contrastive learning, but it is necessary to explain in a few lines what are the performance goals of the proposed framework and how these critical points are overcome.

The discussion of contrastive self-supervised learning and the scheme in Fig. 2 should be moved to a later section where preliminary methods and techniques are presented.

The flow chart in Fig. 3 is too simplified. It should represent all the functional components of a chip solder joint defects detection method and the relationships between them.

Also, to avoid confusion, the framework modeled in Fig. 13 should not be discussed in the same section, but in a later section.

Authors should revise section 2 by adapting it to a more structured way in which each of the functional components in the flow chart in figure 2 is labeled and described; the proposed framework depicted in Fig. 13 is to be described in a later section, so that section 2 becomes a preliminary section.

The performance comparison results, shown in tables 7 and 8, need to be discussed in more detail. For example, regarding the Chip-A Dataset, the OK recall and NG precision obtained by CPC-v2 are higher than those obtained with 0.25-DFN-MobileNetV2. Authors must explain these results and not limit themselves to explaining that 0.25-MobileNetV2 provides a higher F1-score on NG data than that obtained with the other methods.

Reviewer 3 Report

1. Line 266.“ ...the learning rate was 0.00002.” What does it mean? 0.0002 sec-1? Some comments are desirable.

Round 2

Reviewer 1 Report

All my concerns have been addressed. I recommend this paper for publication. 

Reviewer 2 Report

Authors took into account all my suggestions, highlighting more the motivations and objectives of the research and deepening the description of the proposed framework and the discussion of the results of the comparative tests. I consider this paper publishable in the current form.